# SWE-Perf: Can Language Models Optimize Code Performance on Real-World Repositories?

Xinyi He [1]   Qian Liu [2]   Mingzhe Du [3]   Lin Yan [2]   Zhijie Fan [2]
Yiming Huang [4]   Yin Zheng [2]   Zejian Yuan [1]   Zejun Ma [2]

## Abstract

Code performance optimization is paramount in real-world software engineering and critical for production-level systems. While Large Language Models (LLMs) have demonstrated impressive capabilities in code generation and bug fixing, their proficiency in enhancing code performance at the repository level remains largely unexplored. To address this gap, we introduce SWE-Perf, the first benchmark specifically designed to systematically evaluate LLMs on code performance optimization tasks within authentic repository contexts. SWE-Perf comprises 140 carefully curated instances, each derived from performance-improving pull requests from popular GitHub repositories. Each benchmark instance includes the relevant codebase, target functions, performance-related tests, expert-authored patches, and executable environments. Through a comprehensive evaluation of representative methods that span file-level and repo-level approaches (e.g., Agentless and OpenHands), we reveal a substantial capability gap between existing LLMs and expert-level optimization performance, highlighting critical research opportunities in this emerging field.

## 1. Introduction

Recent advances in Large Language Models (LLMs) have significantly enhanced automated code generation and software development assistance, exemplified by tools like GitHub Copilot (Microsoft, 2025) and Cursor (Cursor, 2025). This progress has spurred growing interest in repository-level software engineering challenges in real-world settings (Jimenez et al., 2024). Recent work has introduced multiple benchmarks for evaluating LLM code correctness, including SWE-Bench (Jimenez et al., 2024) and SWE-Dev (Du et al., 2025), as well as frameworks such as AgentLess (Xia et al., 2024) and OpenHands (Wang et al., 2024) that aim to improve the accuracy of LLM-generated code. However, while correctness is foundational in production, performance optimization often yields more profound system-wide benefits (Nascimento et al., 2023; Mancebo et al., 2021; Pereira et al., 2021). Performance optimization is a task traditionally requiring specialized human expertise and poses significant challenges for LLMs. This raises a critical research question: **Can language models effectively optimize code performance in real-world repositories?**

While software engineering and code performance optimization have progressed significantly as distinct fields, current benchmarks struggle to evaluate tasks demanding their integration, especially for repository-level code performance optimization. Repository-level software engineering benchmarks (Guo et al., 2024; Jimenez et al., 2024; Mündler et al., 2024; Du et al., 2025; Miserendino et al., 2025) face particular challenges in supporting open-source code performance optimization. Evaluating whether LLMs can achieve meaningful optimizations is hindered by the lack of human reference implementations for "optimal" efficiency, making it unclear whether code can be improved further. Meanwhile, existing benchmarks focused on code performance (Du et al., 2024; Huang et al., 2024; Liu et al., 2024) primarily target function-level optimizations for algorithmic problems, consequently neglecting the complexities of repository-scale optimization. This represents a key departure from real-world practice, where collaborative improvements between files and modules typically unlock far greater optimization potential than isolated function-level changes.

To address these gaps, we propose **SWE-Perf**, a new benchmark to evaluate the ability of LLMs to optimize code performance in real-world repository-level software engineering scenarios. As shown in Figure 1, the objective is to optimize the performance of a given repository codebase in the context of target functions. Each sample produces a patch that is applied to the original codebase, and the modified codebase is subsequently evaluated for performance improvements.

---

[1]Xi'an Jiaotong University [2]TikTok [3]National University of Singapore [4]University of California San Diego. Correspondence to: Qian Liu <liuqian.sea@gmail.com>.

*Proceedings of the 43rd International Conference on Machine Learning*, Seoul, South Korea. PMLR 306, 2026. Copyright 2026 by the author(s).

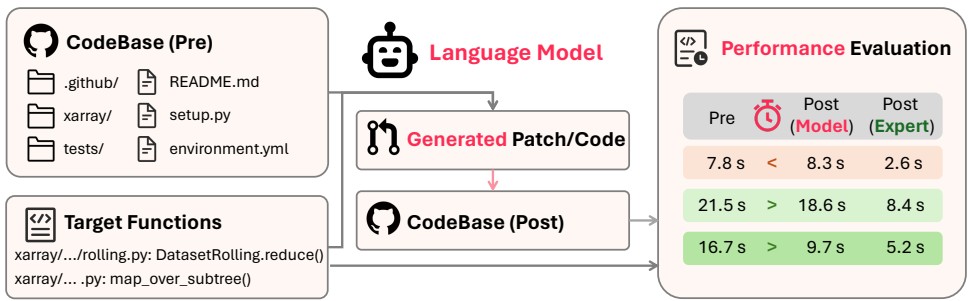

*Figure 1.* The workflow of our SWE-Perf benchmark which evaluates code performance optimization capabilities of language models. The benchmark evaluates language models by providing source code and performance-related tests, challenging them to generate optimized patches. Model performance is evaluated by the runtime gains on the tests, with expert performance as reference.

To construct SWE-Perf, we first extracted pull requests (PRs) from popular GitHub repositories, selecting those with strong indications of performance optimization potential. We then rigorously filtered 100K PRs by evaluating unit test runtimes on both pre-patch (original) and post-patch (modified) codebases. This process resulted in 140 data examples exhibiting observable and stable performance gains, drawn from 9 widely used GitHub repositories. Each example comprises the repository codebase, target functions, performance-related tests, the expert-authored patch, an executable environment (e.g., Docker image), and all runtime metrics. Crucially, the expert patch serves not only to confirm the feasibility of improvement but also as a human-derived gold standard against which to evaluate LLM-generated code performance optimization edits.

We evaluated several leading LLMs under two experimental settings: *oracle (file-level)* and *realistic (repo-level)*. The oracle setting assesses whether a model can optimize code performance when given relevant file contexts, whereas the realistic setting evaluates whether a model can serve as an autonomous agent capable of navigating and operating within the entire repository context without constraints. The experimental results indicate that, relative to expert performance, all LLMs exhibit significant code performance gaps, highlighting opportunities for further advancement. Furthermore, by comparing expert and model results, we analyzed the characteristics and limitations of models in handling performance enhancement tasks, thereby providing insights for further research on improving model performance. In summary, our main contributions are as follows:

- We collect the SWE-Perf dataset, the first benchmark designed to evaluate the ability of language models to optimize code performance on real-world repositories.
- We propose a repository-level performance optimization data collection pipeline, which includes a systematic method for data collection and a set of identification metrics. This framework can be easily extended to other repository-level software engineering datasets.
- We design evaluation metrics specifically for performance optimization and conduct evaluations on several LLMs

under two settings. Our experimental results highlight the unresolved challenges to meet the practical requirements of performance-aware code optimization.

## 2. Related Work

**Code Efficiency**  Recent datasets targeting code efficiency, including Mercury (Du et al., 2024), EFFIBENCH (Huang et al., 2024), EvalPerf (Liu et al., 2024), PIE (Shypula et al., 2024) and KernelBench (Ouyang et al., 2025), primarily focus on function-level performance optimization. While valuable for isolated evaluation, these approaches overlook the complexity of real-world efficiency challenges that span multiple files and modules. This oversimplification limits their ability to benchmark models' capabilities in addressing cross-cutting concerns such as dataflow refactoring or parallelism, where optimization potential is typically more substantial.

**Repository-Level SWE Tasks**  SWE-Bench (Jimenez et al., 2024) first introduced a benchmark for repository-level software engineering tasks. Subsequently, related datasets including SWE-Gym (Pan et al., 2024), SWE-Lancer (Miserendino et al., 2025), SWE-Flow (Zhang et al., 2025), and SWE-Dev (Du et al., 2025) were developed to support various purposes, including model training and unit test evaluation. However, these datasets are primarily tailored for tasks with well-defined objectives, such as bug fixing. In contrast, efficiency optimization represents an open-ended problem lacking standardized solutions. This introduces additional complexities, including identifying optimization targets, designing performance-oriented changes, and requiring long-context understanding, planning capabilities, and efficiency-specific domain knowledge, capacities not fully addressed by existing datasets. A concurrent work, GSO (Shetty et al., 2025), identifies performance-improving commits by combining an LLM-based judge with code-change heuristics, whereas our approach leverages pull requests and runtime environments for identification.

**Approaches to SWE Tasks** To tackle repository-level tasks, prior work has explored two major paradigms: pipeline-based, and agent-based approaches. Pipeline-based techniques, such as Agentless (Xia et al., 2024), SWE-Fixer (Xie et al., 2025), use staged workflows to solve SWE problem. Agent-based systems, like Open-Hands (Wang et al., 2024), SWE-Agent (Yang et al., 2024), SWE-Smith (Yang et al., 2025) and SWE-Search (Antoniades et al., 2025) enable iterative reasoning across multiple steps via autonomous agents. However, these methods were not designed for open-ended performance optimization, leaving room for adaptation and further exploration.

## 3. SWE-Perf Dataset

The SWE-Perf dataset is constructed by collecting data with performance improvement potential from popular GitHub repositories. It is designed to evaluate the capability of LLMs to optimize the performance of real-world software repositories. In the following, we provide detailed descriptions of task formulation (§3.1), data collection (§3.2), and data statistics and distribution (§3.3).

### 3.1. Task Formulation

As illustrated in Figure 1, the input to the SWE-Perf task consists of a codebase and a set of target functions. The output is a performance optimization patch or code, which can be applied to the original codebase to generate a new codebase.

The incorporation of target functions into task inputs is employed to restrict the evaluation scope to performance-related tests. This approach is motivated by the following considerations:

1. Running the full set of unit tests for an entire repository can be prohibitively time-consuming, which hinders the practical applicability of the benchmark. For example, in the case of the xarray repository, there are on average over 220,000 test cases, and testing just a single sample can take over one hour (on a single-core CPU).

2. Given the large codebase of most repositories, there are potentially many locations where performance can be improved. Without targeted guidance, identifying and optimizing relevant regions poses significant challenges for both the model and the evaluation process.

However, as language models continue to advance and become more capable of handling full-repository optimization, we encourage future work to explore approaches that omit the target functions and instead directly optimize the entire codebase.

### 3.2. Data Collection

Figure 2 illustrates the data collection process, and the specific steps of each phase are as follows:

**Phase 1: Collect Pull Requests (PRs).** This phase is conducted by following the methodologies of SWE-bench (Jimenez et al., 2024) and SWE-GYM (Pan et al., 2024). First, we collect high-star, popular GitHub repositories. Specifically, we adopt the same 12 repositories used in SWE-bench. Second, we crawl PRs from these popular repositories. As our subsequent filtering criteria differ from those used in SWE-bench, in order to obtain more data, we re-crawl the aforementioned repositories. Third, we apply attribute filtering. In SWE-GYM and SWE-bench, two main filtering criteria are used: (1) Resolves an issue, and (2) Contributes tests. In this work, we retain only the first criterion. The second criterion is not adopted because our focus lies in whether the PR affects performance, specifically execution time, rather than whether it changes the correctness of unit tests. Therefore, we allow PRs that do not contribute tests, that is, PRs that do not modify unit tests.

**Phase 2: Measure CodeBases Performance.** Each PR collected in Phase 1 contains original and modified codebases. In this phase, we evaluate the performance of all unit tests contained in these codebases. The goal is to identify PRs that lead to performance improvements.

1. Build environment. To ensure correct execution of each codebase, we follow the approach in SWE-GYM to build a corresponding Docker image and executable Docker container for each codebase. Codebases that fail to build successfully are excluded. To ensure the comparability of performance measurements, we constrain each container to a single CPU core and 16 GB memory (5 CPU cores in Phase 4 and evaluation).

2. Execute unit tests. For each codebase, we run all unit tests inside its corresponding Docker container using pytest. This is the most time-consuming step in the entire data collection pipeline. First, the total number of codebases is large. By this stage, we have collected a total of 25,264 codebases. Second, each codebase often contains a large number of unit tests. For example, in the xarray repository, each codebase contains, on average, over 220,000 test cases, and testing a single codebase may take over one hour on a single-core CPU.

3. Record Runtimes. We use pytest to collect the execution time (runtime) of each unit test within the codebase. These runtimes serve as the basis for identifying performance changes between original and modified codebases in the subsequent phase. Codebases for which runtimes cannot be successfully collected are excluded from further analysis.

To minimize environmental effects and ensure comparability

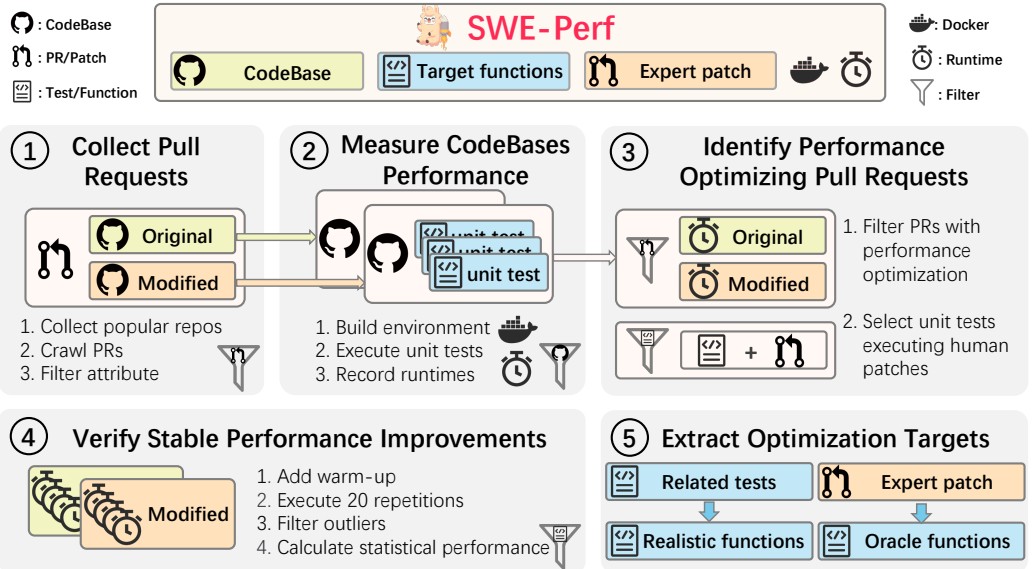

*Figure 2.* The data collection pipeline of our SWE-Perf benchmark. (1) collecting pull requests from popular repositories, (2) measuring unit-test performance of original and modified codebases, (3) identifying performance-optimizing pull requests, (4) verifying stable performance improvements via statistical testing, and (5) extracting optimization targets for both Oracle and Realistic settings.

of performance measurements across codebases, we took two specific steps in this phase: 1. We standardized the execution environment by limiting each Docker container to use one CPU core and 16 GB of memory. 2. Each codebase was evaluated in three repeated experimental runs, producing three runtime measurements per unit test, which are subsequently used in Phase 3.

**Phase 3: Identify PRs with Performance Optimizing Pull Requests.** Based on the performance data collected in Phase 2, we identified pull requests that demonstrate significant performance improvements attributable to the associated code modifications.

1. Filter PRs with performance optimization. After Phase 2, for each PR, we have the performance (i.e., runtimes) of all unit tests in both the original and modified codebases, with each unit test measured in three repeated runs. Our goal is to identify significantly improved unit tests. PRs that do not contain such unit tests are discarded. There are two filtering criteria:

(1) Correctness: The unit test must pass in both the original and modified codebases, i.e., the pytest result must be "pass" in both cases.

(2) Performance Ratio: To ensure that the improvement is substantial in practical terms, we compute the optimized ratio of the modified to original runtimes. The optimized ratio is computed per unit test per pull request as the mean of three experimental replicates. The ratio must exceed a specified threshold (0.3). The ratio is calculated as:

$$Ratio = \frac{R_{original} - R_{modified}}{R_{original}}$$

where $R$ is the runtime for each unit test.

2. Select unit tests executing human patches. To ensure performance improvements are attributable to the associated code modifications, we dynamically execute unit tests for each screened PR. We identify unit tests that, during dynamic execution: (1) exercise the patched code segments modified in the PR, and (2) do not execute any unit tests modified within the PR.

**Phase 4: Verify Stable Performance Improvements.** Following the initial screening, we obtain a preliminary dataset. To ensure stable performance improvements, we verify results using the following procedure, retaining only unit tests whose performance gain exceeds a predefined threshold:

1. Add warm-up. To mitigate initialization-induced timing inaccuracies, before each performance measurement, we execute three performance-related unit tests to warm up the environment.

2. Execute 20 repetitions. Each unit test is run 20 times to ensure runtime stability.

3. Filter outliers. Runtime outliers within the 20 measurements are identified and removed using the Interquartile Range (IQR) method with a threshold multiplier of 1. Specifically, let $Q1$ and $Q3$ represent the first and third quartiles of the runtime sample, respectively. The interquartile range is $IQR = Q3 - Q1$. Data points $r_i$ satisfying either condition below are classified as outliers and removed:

$$r_i < Q1 - k \times IQR, \ r_i > Q3 - k \times IQR \qquad (1)$$

where, $k = 1$ (the threshold multiplier).

4. Calculate statistical performance. To confirm stable per-

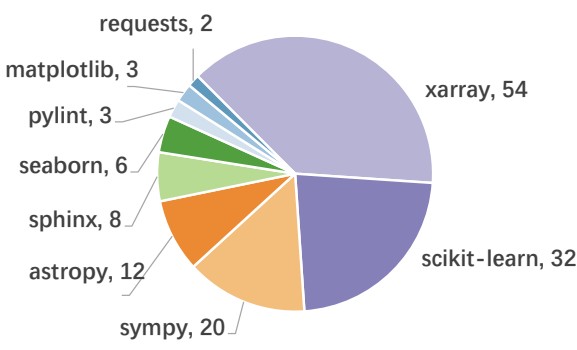

*Figure 3.* Distribution of SWE-Perf across 9 open source popular GitHub repositories.

*Table 1.* Average and maximum values of various SWE-Perf attributes.

| Category | Metric | Mean | Max |
|---|---|---|---|
| Size | # Instances | 140 | |
| | # Repos | 9 | |
| Codebase | # Files (non-test) | 447.3 | 1,972 |
| | # Lines (non-test) | 170k | 502k |
| Expert Patch | # Lines edited | 131.1 | 1,967 |
| | # Files edited | 4.3 | 71 |
| | # Func. Edited | 7.6 | 94 |
| Tests | # Related tests | 8.1 | 68 |
| | Original runtime / $s$ | 0.28 | 25.2 |
| Functions | # Oracle | 7.6 | 94 |
| | # Realistic | 30.1 | 256 |
| Performance | Ratio | 10.9% | 87.8% |

---

**Algorithm 1** Compute Statistically Significant Minimum Performance Gain ($\delta$)

---

**Input:**
  $A = [a_1, a_2, \ldots, a_n]$ : Filtered runtimes (modified version)
  $B = [b_1, b_2, \ldots, b_m]$ : Filtered runtimes (original version)
  $\alpha$ : Significance level (default = 0.1)
  $step$ : Gain increment step (default = 0.01)
  $max_x$ : Maximum gain to test (default = 1.0)
**Output:**
  $\delta$ : Conservative minimum significant performance gain
$x \leftarrow 0.0$
$\delta \leftarrow 0.0$
**while** $x \leq max_x$ **do**
  $B_{\text{adj}} \leftarrow B \times (1 - x)$ // Pessimistically weaken improvement
  $p \leftarrow \text{MannWhitneyUTest}(B_{\text{adj}}, A, \text{alternative} = \text{'greater'})$
  **if** $p < \alpha$ **then**
    $\delta \leftarrow x$                        // Update conservative gain
    $x \leftarrow x + step$                  // Test next larger gain
  **else**
    **break**                      // Significance lost, stop searching
  **end if**
**end while**
**return** $\delta$

---

formance improvement, we compute a statistically significant minimum performance gain ($\delta$) for each test case using Algorithm 1. Here, $\delta$ denotes the largest value such that the modified runtime distribution remains significantly faster than the original under conservative adjustments (Mann-Whitney U test, $p < p\_threshold$ (0.1). Unit tests with $\delta$ exceeding the threshold (0.05) constitute the final SWE-Perf dataset.

**Phase 5: Extract Optimization Targets.** Following the aforementioned four phases, we have filtered PR data and associated unit tests that demonstrate stable performance improvements. In this phase, we extract the target functions for model optimization. We categorize the task into two settings and extract target functions accordingly: Oracle and Realistic.

1. Oracle (File-Level): This setting aims to provide the model with the oracle functions as the optimization target and the related entire files as contextual information, evaluating the model's capability to generate purely performance-enhancing code. The target functions are directly modified. We extract this target functions from the human patch in the PR by combining AST (Abstract Syntax Tree) analysis with unified diff matching.

2. Realistic (Repo-Level): This setting simulates an end-to-end real-world scenario, providing the system (e.g., Open-Hands Agent) with the functions measured during testing (unit test execution) as the optimization target. The system has greater freedom to modify code across the entire repository, measuring the system's ability to enhance performance repository-wide. The target functions are the directly measured functions, not necessarily the ones directly modified, as improvements may involve functions they call. Compared to the Oracle setting, the Realistic setting is more challenging, requiring both performance-improving code generation and additional capabilities such as retrieval.

We identify the target functions by using yappi to record functions dynamically executed during performance-related unit tests. Combined with AST parsing, we determine the specific functions directly invoked by the unit test. We avoid using the unit test itself as the target to prevent test information leakage, which could lead the model to perform functional pruning – modifying the code to retain only the functionality exercised by the test and solely to meet the optimization metric.

After the aforementioned five phases, we transform them into the SWE-Perf dataset, which consists of the following components:

1. **CodeBase**: The source code from the original codebase.

2. **Executable Environment**: The Docker image and container used to execute the codebase.

*Table 2.* Experimental results of leading LLMs under Oracle and Realistic settings on SWE-Perf.

| Setting | Methods | Apply | Correctness | Performance |
|---|---|---|---|---|
| | Expert | 100.00% | 100.00% | 10.85% |
| Oracle (File-Level) | Claude-3.7-sonnet | 66.43% | 61.43% | 1.24% |
| | Claude-4-sonnet | 73.57% | 70.00% | **1.76%** |
| | Claude-4-opus | 85.71% | 78.57% | 1.28% |
| | GPT-4o | 63.57% | 56.43% | 0.60% |
| | OpenAI-o1 | 66.42% | 63.57% | 0.41% |
| | OpenAI-o3 | 78.57% | 76.43% | 1.37% |
| | DeepSeek-V3 | 47.85% | 42.86% | 0.54% |
| | DeepSeek-R1 | 55.71% | 51.43% | 0.90% |
| | Gemini-2.5-Pro | **95.00%** | **83.57%** | 1.48% |
| | Qwen3-235B-A22B | 54.29% | 48.57% | 0.68% |
| Realistic (Repo-Level) | Claude-3.7-sonnet (Agentless) | **88.57%** | 70.71% | 0.41% |
| | Claude-3.7-sonnet (OpenHands) | 87.86% | **77.86%** | **2.26%** |

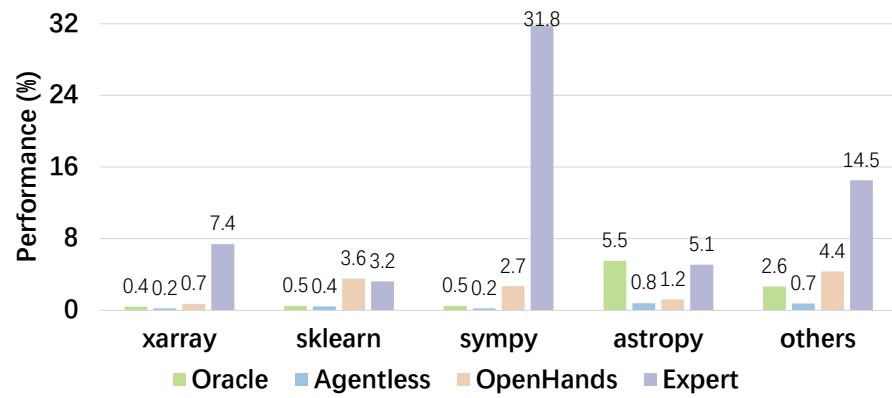

*Figure 4.* Performance for different methods across the 9 repositories represented in SWE-Perf. The base model used in the Oracle setting is Claude-3.7-sonnet.

3. **Target Functions**: The optimization-target functions categorized as oracle and realistic.

4. **Performance-related Unit Tests**: The unit tests identified as having performance improvements.

5. **Runtime Metrics**: Original and modified codebase runtime metrics for performance-related tests.

6. **Expert Patch**: The expert patch from the modified codebase. These serve as references for human-level performance optimization.

### 3.3. Data Statistics and Distribution

In Phase 1, a total of 102,241 pull requests were collected, from which 19,797 pull requests remained after filtering. In Phase 2, 34,397 distinct codebases were gathered, and test executions were successfully performed on 19,499 of them, yielding corresponding runtime data. In Phase 3, 4413 PRs were identified whose main and dev codebases both had available runtime data, from which 1,696 valid instances were derived. In Phase 4, 140 instances were identified. Detailed statistics and data distributions of SWE-Perf are presented in Figure 3 and Table 1.

## 4. Evaluation Methodology

We designed a three-level performance evaluation framework with three progressively stringent metrics: *Apply*, *Correctness*, and *Performance*. During evaluation, we first apply the model-generated patch/code to the original codebase ($codebase\_pre_i$) to obtain the post-patch codebase ($codebase\_post_i$). Subsequently, within the corresponding Docker environment, we execute all performance-related tests ($test_{i,j}, j \in \{1, \ldots, N_i\}$) on both the original and post-patch codebases, collecting the results ($result\_pre_{i,j}, result\_post_{i,j}$) and runtime ($runtime\_pre_{i,j}, runtime\_post_{i,j}$) measurements for comparison.

It is worth noting that, to eliminate the impact of environmental variability on execution speed, we re-evaluated the original codebase runtime during the testing phase, even when the original codebase runtime from data collection was available. This ensures full comparability between the original and post-patch codebase runtime measurements.

The definitions and corresponding formulas for the three evaluation metrics are as follows:

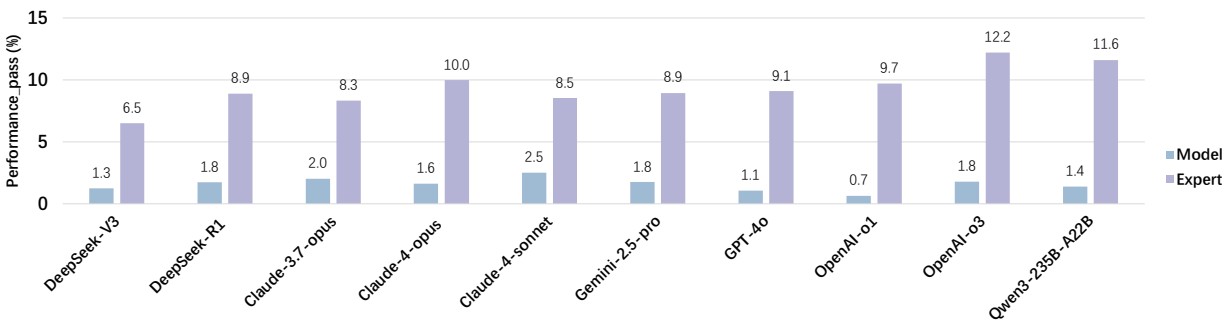

Figure 5. Performance for correct examples. The expert performance calculated using only correct examples from the corresponding method.

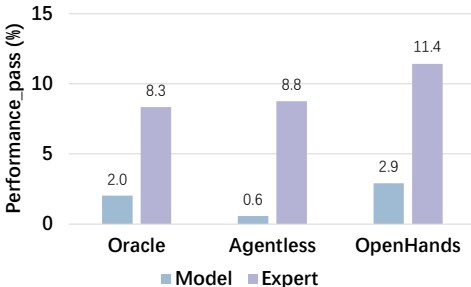

Figure 6. Performance for correct examples. The expert performance calculated using only correct examples from the corresponding method.

**Apply**: This metric evaluates whether the model-generated patches or code can be successfully applied to the original codebase without conflicts or errors. $Apply = \frac{N_{apply}}{N_{total}}$, where, $N_{apply}$ is the number of successfully applied samples, $N_{total}$ is the number of total samples.

**Correctness**: This metric assesses whether the patch preserves the functional correctness of the code, specifically whether all unit tests pass successfully after the patch is applied.

$$Correctness = \frac{\sum_{i=1}^{N_{total}} \left[ \bigwedge_{j=1}^{N_i} result\_post_{i,j} = pass \right]}{N_{total}}$$

where $result\_post_{i,j}$ is the post-patch result of the $j$-th unit test on the $i$-th sample. $N_i$ is the number of performance-related unit tests for the $i$-th sample. $\bigwedge$ is the logical AND for all tests $j$, indicating the sample must pass every test.

**Performance**: This metric measures the statistically significant minimum performance gain introduced by the patch, based on runtime comparisons. The computation process resembles that of Phase 4 in data collection (§3.2). After a warm-up period, 20 repetitions and outliers filtering, the Minimum Performance Gain ($p_{i,j}$) for each instance $i$ and each unit test $j$ is calculated using Algorithm 1. Performance is then calculated as follows:

$$Performance = \frac{1}{N_{total}} \sum_{i=1}^{N_{total}} P_i, \ \ P_i = \frac{1}{N_i} \sum_{j=1}^{N_i} p_{i,j} \quad (2)$$

## 5. Experiments

This section presents the baseline setting (§5.1), the main experimental results (§5.2), and a further analysis of the model's performance (§5.3).

### 5.1. Baselines

Recent work on repository-level software engineering tasks (*e.g.*, SWE-Bench) can be broadly categorized into three paradigms: direct model approaches, pipeline-based methods and agent-based systems. We select representative state-of-the-art methods from each category for evaluation.

1. Oracle: For the oracle setting, we adopt a chain-of-thought prompting strategy to directly use model to enhance codebase performance. The model is provided with the oracle files extracted from expert patches and oracle target functions. A single-pass inference is used to generate the patch. We evaluate with 10 popular models.

2. Agentless (Pipeline-Based): Agentless (Xia et al., 2024) follows a pipeline-based approach to address the task. It employs a fixed multi-stage workflow consisting of hierarchical fault localization, code repair, and candidate patch selection through regression and reproduction testing.

3. OpenHands (Agent-Based): OpenHands (Wang et al., 2024) provides a flexible and extensible platform for building autonomous software development agents, enabling iterative reasoning and interaction across multiple steps in the software engineering process.

For both Agentless and OpenHands, we use Claude-3.7-sonnet as the base model. Because Claude-3.7-sonnet is the officially recommended backend for OpenHands [1], as it has been reported to work best within OpenHands. The implementation details are provided in Appendix §C.

---

[1] https://github.com/All-Hands-AI/OpenHands

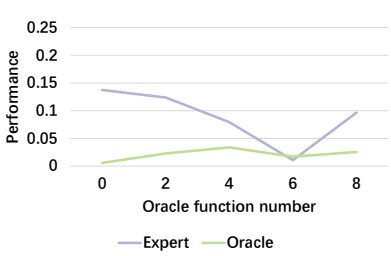

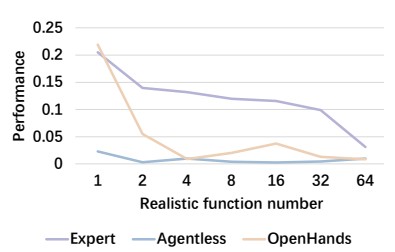

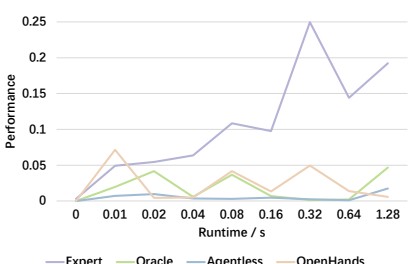

*Figure 7.* Performance variation relative to the number of Oracle target functions.

*Figure 8.* Performance variation relative to the number of Realistic target functions.

*Figure 9.* Performance variation relative to the runtime of the original codebase.

## 5.2. Main Results

The performance results of various methods on SWE-Perf are presented in Table 2. The comparative performance across different repositories is illustrated in Figure 4. From these empirical findings, we derive the following conclusions:

**Compared to the expert, all models exhibit substantial room for improvement on SWE-Perf.** OpenHands performs best, due to its agent-based methodology, providing a flexible and extensible platform for autonomous software development agents that enable iterative reasoning and multi-step interaction throughout the software engineering process. However, a significant performance gap of 8.59% persists between OpenHands and Expert, indicating considerable potential for further.

**The model exhibits the potential to surpass expert performance and achieve superior results.** As can be observed in Figure 4, the model already rivals the performance of the Expert on certain repositories; for instance, on sklearn, OpenHands outperforms the Expert by 0.4%. This demonstrates that the model can achieve a competitive edge over established expert methods in specific tasks or datasets, signifying an early-stage breakthrough.

## 5.3. Analysis of Model Capabilities

This subsection analyzes performance from four aspects: (1) isolating performance from correctness metrics, (2) quantifying target functions' impact, (3) evaluating runtime-performance relationships, and (4) identifying keyword patterns. We aim to uncover optimization bottlenecks and actionable improvement insights.

### 5.3.1. PERFORMANCE ANALYSIS DECOUPLED FROM CORRECTNESS

To decouple the model's code performance enhancement capability from its code/patch generation correctness, we calculated performance metrics exclusively for correct examples by modifying the denominator in Equation (2) from $N_{total}$ to $N_{correctness}$. The results are presented in Fig-

ure 6 and Figure 5. The Expert metric represents expert performance calculated using only correct samples from the corresponding method.

**OpenHands demonstrates superior performance particularly excelling in scenarios with higher potential performance ceilings.** As illustrated in Figure 6, using identical models, OpenHands achieves an approximately 3% higher benchmark performance compared to alternative methods, highlighting its superior capability in translating model capacity into realized performance, especially near the achievable ceiling.

### 5.3.2. IMPACT OF RUNTIME ON PERFORMANCE

To examine the model's ability to improve performance across runtimes, we plotted the chart shown in Figure 9.

**As runtime increases, the performance ceiling rises correspondingly.** As shown in Figure 9, the expert model's performance demonstrates a progressive upward trend with extended runtime, indicating that longer computation times enable more sophisticated optimization and convergence towards higher performance potentials.

**The model's capability to improve performance on cases with longer runtimes requires further enhancement.** Figure 9 reveals that while expert performance continues to climb with increased runtime, the model's performance plateaus (or remains stagnant). This divergence underscores the critical need to analyze and emulate the expert's optimization strategies specifically under extended-runtime conditions to enhance the model's performance scalability.

### 5.3.3. IMPACT OF TARGET FUNCTIONS ON PERFORMANCE

To investigate the impact of the number of target functions on model performance, we present the statistics summarized in Figure 7 and Figure 8.

**As the number of target functions increases, performance improvements become increasingly difficult to achieve.** Figure 8 shows expert performance declines with the addition of functions, indicating heightened difficulty in

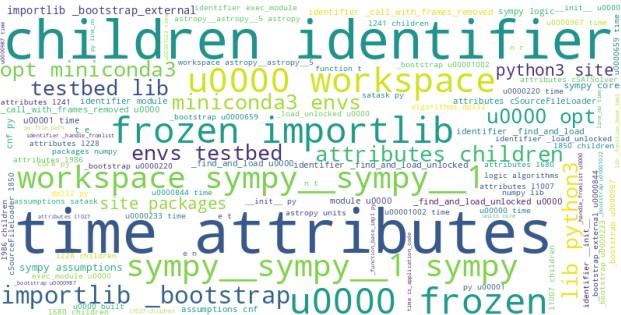

*Figure 10.* Word cloud of lines added in OpenHands patches.

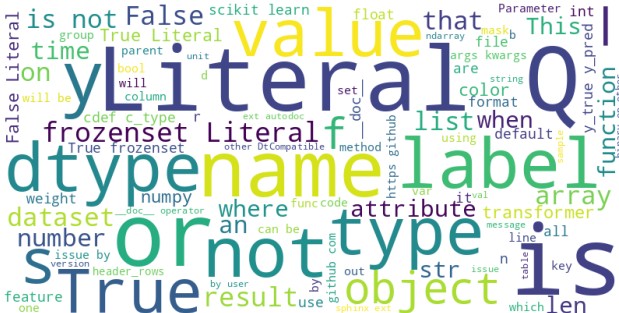

*Figure 11.* Word cloud of lines added in Expert patches.

enhancing performance and a lower performance ceiling.

**The model should prioritize learning from the expert to enhance its capability when handling a larger number of target functions.** Figure 8 reveals that OpenHands achieves performance comparable to the expert at lower function counts. However, the performance gap widens significantly as the number of functions increases. Future research aimed at improving model performance should focus on optimizing for multi-function scenarios.

### 5.3.4. KEYWORD ANALYSIS FOR PERFORMANCE

To further investigate the differences in modification strategies preferred by the model and the expert when enhancing performance, we respectively generated word clouds representing the lines added in patches, as shown in Figure 10 and Figure 11.

**OpenHands patches focus on low-level data structures and basic functionality.** The word cloud reveals frequent terms such as "children," "identifier," "time," and "attributes," indicating that the changes are centered on refining structural components and enhancing fundamental operations. These optimizations likely address data management and attribute handling, focusing on the internal mechanics of the code.

**Expert patches emphasize high-level abstractions and data integrity.** The presence of terms like "literal," "value," "type," "label," and "dtype" suggests that the optimizations are geared towards improving type safety, type annotations, and handling of data values. These changes likely aim to enhance error handling, code clarity, and overall system efficiency by addressing more abstract elements of the code.

**The model-generated patches exhibit a strong focus on foundational infrastructure and low-level operations**, as indicated by frequent terms such as "miniconda3", "envs", "frozen", "importlib", and "bootstrapu0000". This suggests an emphasis on environment configuration, dependency management, and basic module handling. The presence of encoded fragments ("u0000") further implies automated generation targeting syntactic adjustments or toolchain com-

patibility, rather than semantic, application-level optimization.

**The expert patches demonstrate a clear orientation towards domain-specific functionality and performance-critical enhancements.** Key terms like "sympy", "time", "workspace", and "active packages" reveal a deliberate strategy to optimize core computational workflows (e.g., symbolic mathematics with sympy), resource management (workspace), and runtime efficiency (time). This reflects human expertise in restructuring high-level components to improve performance, maintainability, and domain-relevant operations.

## 6. Conclusion

In conclusion, SWE-Perf addresses a critical gap in current benchmarking by providing the first repository-level dataset focused on realistic code performance optimization, a task traditionally reliant on human expertise and largely unexplored in prior LLM evaluations. Our benchmark and comprehensive baseline assessments reveal substantial room for improvement in current models, underscoring the complexity of cross-module and repository-scale optimizations in real-world software. The significant performance gaps observed between existing LLMs and expert-level optimization highlight the need for novel approaches that can handle the intricacies of repository-level performance enhancement. By establishing this new standard and evaluation framework, we pave the way for future research to advance the capabilities of language models in delivering meaningful, performance-aware code enhancements at scale, ultimately bridging the gap between automated code generation and expert-level optimization in production environments.

## Impact Statement

The SWE-Perf dataset is constructed exclusively from publicly accessible software repositories released under open-source licenses that permit the use of their code for research purposes. The selection of repositories for inclusion is based on objective popularity metrics (e.g., stars, forks) to avoid

introducing intentional or unintentional bias toward specific project types, development communities, or contributor demographics. We release the SWE-Perf benchmark, including task instances, data collection and evaluation tools, and experimental results, publicly to facilitate reproducibility and community adoption. To support ongoing improvement and address potential concerns, we will maintain open channels for feedback and collaboration.

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

# Appendix

## A. Limitations

Our work has two primary limitations. First, the current version of SWE-Perf is constructed from a limited set of open-source repositories, future work could expand the dataset to improve coverage and generalizability. Second, while we use human-written patches as ground truth to assess model performance, these may not represent the optimal achievable performance, potentially underestimating the true upper bound of improvement.

## B. Dataset

All experiments were conducted on two Linux machines. Each machine is equipped with 256 logical CPU cores (128 physical cores across 2 sockets, with 2 threads per core) and 2.0 TiB of RAM.

### B.1. Threshold Choices

We select the filtering and verification thresholds to balance statistical reliability with benchmark construction cost. For the Phase 3 performance-ratio threshold, the average coefficient of variation (CV) observed in our runtime measurements is 10.976%. In engineering measurements, a signal-to-noise ratio of approximately 2–3 is commonly used to distinguish real improvement from measurement noise. Therefore, requiring a 30% runtime improvement provides a signal of roughly 2.7 times the observed noise level.

For Phase 4 verification and evaluation, we use 20 repeated measurements because this provides an effective sample size for the Mann-Whitney U test while keeping the benchmark computationally feasible. We set the significance level to $\alpha = 0.1$, which is suitable for limited-sample non-parametric testing. Finally, the $\delta > 0.05$ threshold is used as a minimum practical performance-gain requirement after statistical testing, filtering out improvements that are statistically detectable but too small to be meaningful in practice.

### B.2. Details of Data Statistics

The statistical summary of data volume at each stage is presented in Table 3.

*Table 3.* Table of data volume at each phase.

| Repo | Phase1 | | | Phase2 | | Phase3 | | | Phase4 |
|---|---|---|---|---|---|---|---|---|---|
| | PRs | Tasks | Tasks with versions | Unique codebase | Codebase with pytest report | Tasks with success runtimes | Tasks after step1 | Tasks after step2 | Tasks |
| astropy | 11555 | 1947 | 1947 | 3589 | 2323 | 529 | 409 | 195 | 12 |
| django | 13200 | 4179 | 4179 | 6887 | 6399 | - | - | - | - |
| matplotlib | 18791 | 2576 | 2576 | 4707 | 1648 | 495 | 369 | 163 | 3 |
| seaborn | 1115 | 304 | 304 | 573 | 539 | 199 | 165 | 76 | 6 |
| flask | 2637 | 285 | 262 | 521 | 410 | 62 | 0 | - | - |
| requests | 2507 | 223 | 217 | 410 | 364 | 151 | 105 | 43 | |
| xarray | 4470 | 1359 | 1354 | 2485 | 1000 | 517 | 495 | 327 | 54 |
| pylint | 4553 | 1174 | 911 | 1680 | 1522 | 806 | 772 | 405 | 3 |
| pytest | 6228 | 1203 | 989 | 1860 | 1717 | 583 | 397 | 0 | 2 |
| sklearn | 18079 | 2576 | 2574 | 4721 | 431 | 181 | 181 | 94 | 32 |
| sphinx | 6002 | 1507 | 1385 | 2629 | 2299 | 630 | 573 | 236 | 8 |
| sympy | 13104 | 2464 | 2464 | 4335 | 847 | 260 | 254 | 157 | 20 |
| sum | 102241 | 19797 | 19162 | 34397 | 19499 | 4413 | 3720 | 1696 | 140 |

Table 4 presents the runtime statistics for each codebase in Phase 2 - Step 2.

The mapping table of repository abbreviations to their full names is provided in Table 5.

### B.3. Coverage and Representativeness

Beyond repository-level statistics, we further characterize the coverage of SWE-Perf by categorizing each instance according to its primary optimization strategy. The benchmark covers four common forms of real-world performance improvement:

*Table 4.* Runtime statistics for each codebase in phase 2 - step 2. Only runtimes from successful pytest executions are included. All time measurements are in minutes. **Test runtime** refers to the time taken to execute pytest for an individual codebase, while **Total duration** denotes the overall execution time for a single codebase, including operations such as Docker image building and pytest execution.

| Repo | Test runtime | | Total duration | |
|---|---|---|---|---|
| | avg | max | avg | max |
| astropy | 3.09 | 21.57 | 4.36 | 24.12 |
| django | 0.22 | 0.91 | 1.52 | 5.85 |
| matplotlib | 7.60 | 16.96 | 10.69 | 28.75 |
| seaborn | 3.37 | 14.56 | 4.15 | 15.97 |
| flask | 0.04 | 0.10 | 1.04 | 1.75 |
| requests | 1.09 | 9.69 | 1.88 | 11.08 |
| xarray | 58.11 | 119.95 | 58.52 | 121.03 |
| pylint | 2.75 | 3.99 | 3.47 | 6.13 |
| pytest | 2.34 | 18.94 | 3.13 | 20.10 |
| sklearn | 83.89 | 119.96 | 85.83 | 125.98 |
| sphinx | 8.72 | 38.99 | 9.57 | 40.00 |
| sympy | 24.60 | 112.17 | 24.98 | 112.57 |

*Table 5.* Mapping table of repository abbreviations to full names.

| Repo name | Repo full name |
|---|---|
| astropy | astropy/astropy |
| matplotlib | matplotlib/matplotlib |
| seaborn | mwaskom/seaborn |
| requests | psf/requests |
| xarray | pydata/xarray |
| pylint | pylint-dev/pylint |
| sklearn | scikit-learn/scikit-learn |
| sphinx | sphinx-doc/sphinx |
| sympy | sympy/sympy |

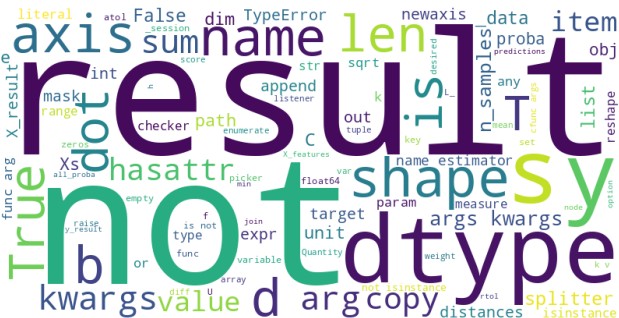

*Figure 12.* Word cloud of lines added in Agentless patches.

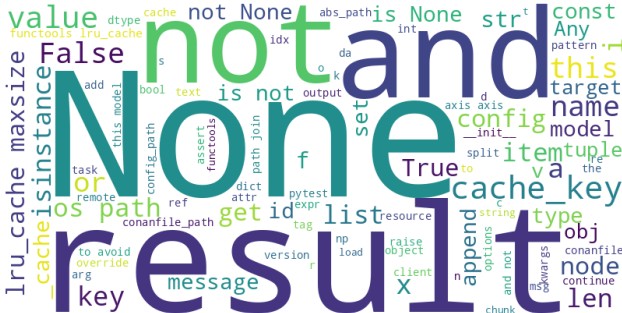

*Figure 13.* Word cloud of lines added in Oracle(Claude-3.7) patches.

- **Redundancy Elimination (25.7%):** eliminating duplicate calls, repeated attribute lookups, and invalid branches.
- **Algorithmic Logic (26.4%):** improving computational efficiency through vectorization, formula simplification, and complexity reduction.
- **Data & Resource Optimization (15.0%):** improving dtype management, lazy loading, and memory layout.
- **Cache & Compatibility (32.9%):** reusing state through caching and adapting logic to performance-sensitive backends such as NumPy.

This distribution suggests that SWE-Perf is not limited to a single optimization pattern, although its current coverage remains concentrated in Python repositories and should be expanded to broader project domains and programming languages in future versions.

## C. Baselines

### C.1. Details of Baselines

1. Oracle: The specific prompts used for the Oracle is shown in Figure 14.

   (a) OpenAI/GPT: The model version used is o1-preview-2024-09-12, o3-2025-04-16, gpt-4o-2024-11-20, with a temperature of 0.2, top-p of 0.1, and a maximum token limit of 8192.

   (b) Claude: The model version is gcp-claude37-sonnet, gcp-claude4-opus, gcp-claude4-sonnet. The thinking feature is enabled, with a thinking budget of 2000 tokens and a maximum token output of 8192.

   (c) DeepSeek: The versions used are deepseek-r1-0528 and DeepSeek-V3.

   (d) Gemini: The versions used are gemini-2.5-pro-preview-05-06.

   (e) Qwen: The versions used are Qwen3-235B-A22B.

2. Agentless [2]: The sample number is set to 1.

3. OpenHands [3]: The maximum number of iterations is set to 2000.

### C.2. Additional Word Clouds

More comprehensive word clouds are presented in Figure 12, Figure 13.

---

[2] https://github.com/OpenAutoCoder/Agentless
[3] https://github.com/All-Hands-AI/OpenHands

---

**Oracle prompt:**

---

You will be provided with a partial code base and objective functions. You need to improve the objective function's efficiency and execution speed by editing the code base.
<problem_statement>
Please enhance the computational efficiency and execution speed across the entire repository. The optimization efforts may target one or more objective functions, including but not limited to:
[target_functions]
The following conditions apply:
1. Acceleration of at least one objective function is sufficient for success, as performance evaluations will be conducted collectively on all targeted functions.
2. Optimization may be achieved either directly through modifications to the objective functions or indirectly by improving computationally intensive subroutines upon which they depend.
3. Optimization efforts should prioritize maximal efficiency gains where feasible.
4. All existing unit tests must remain unaltered to preserve functional correctness.
</problem_statement>


[content_of_files]


Please improve its efficiency and execution speed by generate *SEARCH/REPLACE* edits to fix the issue.

Every *SEARCH/REPLACE* edit must use this format:
1. The file path
2. The start of search block: <<<<<<< SEARCH
3. A contiguous chunk of lines to search for in the existing source code
4. The dividing line: =======
5. The lines to replace into the source code
6. The end of the replace block: >>>>>>> REPLACE
7. You can't edit the test case, only the code base.
8. Only use standard python libraries, don't suggest installing any packages.

Here is an example:

```python
### mathweb/flask/app.py
<<<<<<< SEARCH
from flask import Flask
=======
import math
from flask import Flask
>>>>>>> REPLACE
```

Please note that the *SEARCH/REPLACE* edit REQUIRES PROPER INDENTATION. If you would like to add the line 'print(x)', you must fully write that out, with all those spaces before the code!
Wrap the *SEARCH/REPLACE* edit in blocks ```python...```.

---

*Figure 14.* Oracle prompt.

