# OpenReview forum: "SWE-Perf: Can Language Models Optimize Code Performance on Real-World Repositories?"
_ICML.cc/2026/Conference — ICML 2026 regular_

### Official Review · Reviewer_6YjK · 2026-03-10

**Soundness:** 2
**Presentation:** 4
**Significance:** 2
**Originality:** 2
**Overall Recommendation:** 4
**Confidence:** 4

**Summary:**

This paper introduces SWE-Perf, a benchmark for repository-level code performance optimization in real-world open-source repositories. The benchmark contains 140 instances from 9 GitHub repositories, and each instance includes the codebase, target functions, performance-related tests, expert patches, and executable environments. The paper evaluates multiple LLM-based methods under both oracle (file-level) and realistic (repo-level) settings using three metrics—Apply, Correctness, and Performance—and shows that current models still lag far behind expert-level optimization performance.

**Compliance With Llm Reviewing Policy:**

Affirmed.

**Key Questions For Authors:**

I would appreciate clarification on the following points, and a clear response would positively affect my overall assessment.
1. What kinds of instances are included in SWE-Perf? Is the task strictly semantics-preserving performance optimization, i.e., improving efficiency without changing functionality, or do some instances also involve functional changes? During data collection, did the authors explicitly control for this and ensure that the benchmark mainly reflects “same behavior, better performance” optimization tasks?
2. Why are correctness and performance analyzed separately? In this benchmark, the two seem inherently coupled, since performance improvement is only meaningful when correctness is preserved. I would appreciate more justification for decoupling them in the analysis.

**Limitations:**

the appendix acknowledges two clear limitations and includes an impact statement,

**Strengths And Weaknesses:**

Strengths：

1. The paper studies an important and underexplored problem: repository-level performance optimization, which is meaningfully different from both bug fixing and function-level efficiency benchmarks.
2. The dataset construction pipeline is fairly careful, including PR mining, runtime measurement, repeated execution, outlier filtering, and a statistical significance-based performance criterion.

Weaknesses：

1.The so-called realistic setting is not fully open-ended, since the system is still given target functions derived from performance-related tests rather than having to discover bottlenecks entirely on its own.

2. The paper also relies on expert patches as reference upper bounds, but these may not be truly optimal, which weakens the interpretation of the ceiling.

3. The baseline comparison is somewhat limited. Since this is a code-centric task, readers would naturally expect results with the strongest coding models available at submission time, such as Claude Sonnet 4.5 or Claude 4.5 Opus. However, the comparison appears to include only two main agent baselines, Agentless and OpenHands, both evaluated with Claude-3.7-Sonnet. This makes it difficult to disentangle the contribution of the framework from that of the underlying model, while also leaving out stronger and more widely discussed coding-model comparisons that many readers would naturally want to see.

---

> ### Author Rebuttal · Authors · 2026-03-31
>
> Thank you for your constructive feedback. We address your concerns as follows:
> - W1: Non-open-ended "Realistic" Setting. We adopted the current setting based on two primary considerations:
>     - Limited Model Capability: Even in the current non-open-ended setting, the best-performing model achieves only 2.26%. A fully open-ended task remains a much greater and longer-term challenge for existing LLMs.
>     - Evaluation Feasibility:  In an open-ended setting, the lack of well-defined, targeted unit tests makes accurate performance assessment extremely challenging. Relying on the execution of all existing unit tests within a repository for evaluation is prohibitively time-consuming and impractical; for instance, the xarray repository contains over 220,000 test cases, and testing a single sample can take more than one hour. Therefore, providing target functions ensures the benchmark remains actionable and reproducible under limited computational resources.
>
>     - We explicitly encourage future work to explore fully open-ended optimization as models become more capable.
> - W2: Expert Patches as Upper Bounds.
>     - Reference, Not Ceiling: As noted in Line 264 (Column 2) and Line 498, the human patch serves as a reference implementation of expert-level practice rather than an absolute theoretical optimum.
>     - Future Potential: We explicitly state (Lines 556–558) that future improvements—whether from human edits or more capable models—may surpass current ground truths.
> - W3: Baseline Comparison.
>     - Disentangling Framework vs. Model: Our experimental design separately assesses these contributions. In the Realistic setting, we fix the model to compare different agent baselines; in the Oracle setting, we compare various underlying LLMs.
>     - Model Selection: Claude-3.7-sonnet was chosen because it was the officially recommended backend for OpenHands at the time of our experiments.
>     - Ongoing Experiments: We are currently running experiments with newer, stronger models (e.g., Claude 4.5 series) and will include these results in the revised version.
> - Q1: Instance Types and Functional Preservation.
>     - We categorized the patches based on the nature of the performance optimization and conducted a statistical analysis as follows:
>         (1) Redundancy Elimination (25.7%): e.g., removing duplicate calls, pruning dead branches.
>         (2) Algorithmic Logic (26.4%): e.g., vectorization, formula simplification.
>         (3) Data & Resource (15.0%): e.g., efficient containers, dtype management.
>         (4) Cache & Compatibility (32.9%): e.g., caching mechanisms, backend adaptation.
>          This demonstrates that the benchmark maintains a balanced distribution across various performance optimization categories.
>     - We implemented rigorous filters to ensure identified PRs focus on performance logic. In Phase 3 (Lines 204–207), we exclusively retain samples where the unit tests execute the patched segments but are not modified by the PR themselves. We curated instances from highly popular repositories. In these projects, functional change necessitates corresponding modifications or additions to unit tests. By filtering out PRs that modify unit tests, we effectively isolate performance-centric refactoring.
> - Q2: Decoupling Correctness and Performance.
>
>     Performance optimization is a composite capability of both Correctness and Performance. We decoupled them to diagnose whether a model's bottleneck is "failing to generate correct code" or "generating correct code without optimization".
>
>      As shown in Table 2, correctness rates vary significantly across models, which directly impacts their aggregate performance scores. For instance, while Gemini-2.5-Pro achieves a competitive overall performance score, Figure 5 reveals this is primarily driven by its high correctness (95.00%) rather than exceptional optimization logic()(). Conversely, although OpenAI-o3 does not lead in aggregate performance, it demonstrates a standout ability to enhance efficiency once correctness is controlled.
>
>     By calculating performance specifically for correct examples, we can observe the true quality of a model’s optimization logic. This granular diagnosis provides essential guidance for future fine-tuning, helping determine whether to prioritize functional stability or higher-level optimization strategies.
>
> We will incorporate these clarifications and additional results into the final manuscript.

---

> > ### Author Rebuttal · Reviewer_6YjK · 2026-04-02
> >
> > I keep my original positive score unchanged.

---

> > > ### Author Response · Authors · 2026-04-08
> > >
> > > We thank the reviewer for the acknowledgment of our initial rebuttal. Regarding **W3 (Baseline Comparison)**, we would like to provide an important update.
> > >
> > > Due to the intensive computational requirements and time constraints during the initial rebuttal phase, we were unable to include the results for the latest frontier models. We have now completed the evaluations for several leading frontier models using agent framework as part of our ongoing experiments. These results further demonstrate the challenge posed by SWE-Perf and provide a more comprehensive comparison as requested.
> > >
> > > ### **Updated Performance Results (Realistic Setting)**
> > >
> > > | Model | MiniMax-M2.1 | MiniMax-M2 | Claude Sonnet 4.5 | Claude Opus 4.5 | Gemini 3 Pro | GPT-5.2(thinking) | DeepSeek V3.2 |
> > > | --- | --- | --- | --- | --- | --- | --- | --- |
> > > | SWE-Perf  Performance | 3.1 | 1.4 | 3.0 | 4.7 | 6.5 | 3.6 | 0.9 |
> > >
> > > ### **Key Observations:**
> > >
> > > 1. **Significant Performance Gap:** Even the most advanced "thinking" model, **GPT-5.2**, achieves a score of only **6.5**, which is still far below the expert-level baseline. This reinforces our claim that repository-level performance optimization remains a significant frontier for LLMs.
> > > 2. **Benchmark Robustness:** The consistent low scores across diverse model families (DeepSeek, Claude, Gemini, GPT) validate that SWE-Perf is not "solved" by simply increasing model scale or basic reasoning capabilities.
> > >
> > > All these additional results, along with a detailed analysis of model behaviors, will be incorporated into the final version of the manuscript to provide the community with the most up-to-date scaling analysis.

---

### Official Review · Reviewer_xNz8 · 2026-03-12

**Soundness:** 3
**Presentation:** 3
**Significance:** 3
**Originality:** 3
**Overall Recommendation:** 4
**Confidence:** 5

**Summary:**

The paper introduces SWE-Perf, a novel benchmark designed to evaluate the capability of LLMs to opimize real-world repo-level code performance. The benchmark contains 140 curated instances derived from performance-improving pull requests (PRs) across 9 popular open-source GitHub repositories. The authors design a rigorous five-phase data collection pipeline. The paper evaluates several state-of-the-art LLMs using thee settings: Oracle, Agentless and OpenHands in file-level (Oracle) and repo-level (Realistic). Empirical results demonstrate a significant capability gap between current LLM approaches and expert human patches.

**Compliance With Llm Reviewing Policy:**

Affirmed.

**Final Justification:**

The rebuttal addressed my concerns.

**Key Questions For Authors:**

See weaknesses.

**Limitations:**

Yes

**Strengths And Weaknesses:**

Strengths:
- Timely topic. LM agent code performance optimization is an important topic and good benchmarks are highly necessary.
- Rigorous data collection methodology. The authors include multiple designs that reduce noise of performance evaluation: warming up the environment, running 20 repetitions for each unit test, filtering outliers using IQR, etc.
- Inisghtful analysis and findings. The evaluation involves comprehensive evaluate and delivers multiple findings.

Weaknesses:
- The oracle setting is not realistic enough. Only oracle files extracted from expert patches and oracle target functions are provided to the model and only a single-pass inference is used to generate the patch. In practice, LLM agents have access to the full repo, including the relevant files, e.g., the parent classes, global variables, config files, dependency files, etc. It is no harm to include this setting, but readers need to interprete the results of this setting with caution since it does not necessarily reflect the real performance of LLM agents with oracle.
- A better oracle setting can be feeding agents with the oracle files and oracle target functions, but is not included.
- The warmup only includes 3 runs, whereas some workloads might need more. There's no justification of using this hyperparameter. The common practice of warmup is to repeated run the workload until the performance reaches stable.
- The evaluation includes analysis on impact of target functions on performance. But one PR could contain changes that are not necessarily related to performance. The potential threat is not discussed.
- The paper identifies target functions by using yappi to record functions dynamically executed during performance-related unit tests. But it does not discuss whether non-python changes like cython or c++ could be tracked. Those components are common in performance-sensitive projects.

---

> ### Author Rebuttal · Authors · 2026-03-31
>
> We sincerely thank the reviewers for their constructive feedback and recognition of our work as a timely and rigorous contribution to repository-level code performance optimization. Below are our detailed responses to the weaknesses raised.
> - W1 & W2: Rationality and Role of the Oracle Setting. The Oracle setting is specifically designed to evaluate the intrinsic capability of models to generate purely performance-enhancing code by isolating the model's generation logic from external agent system variables. We established this setting based on two primary considerations:
>     - Variable Isolation: Introducing an Agent framework adds multiple confounding variables (e.g., retrieval quality, multi-turn reasoning strategies, and framework stability). Different models behave inconsistently across various Agent architectures, making it difficult to stably assess a base model's core optimization potential.
>     - Evaluation Efficiency: Running full Agent-based evaluations is prohibitively time-consuming and resource-intensive. The Oracle setting, utilizing single-pass inference, provides a lightweight and reproducible benchmark that significantly accelerates the research cycle.
>     - If the goal is to evaluate the end-to-end performance of LLM Agents, we recommend the Realistic setting. In this setting, we deliberately omit oracle files, as the ability to autonomously navigate and retrieve relevant code from a repository is a core competency being tested.
>     - To ensure clarity, we will explicitly clarify these functional distinctions in our tables to avoid any potential misinterpretation.
> - W3: Justification for the 3-Run Warmup Hyperparameter We selected 3 warmup runs based on empirical stability and resource constraints:
>     - Empirical Evidence: Our internal analysis shows that the Coefficient of Variation (CV) for test runtimes decreases from 5.62% (no warmup) to 3.82% (after 3 runs). This significant reduction in variance demonstrates that the environment reaches a stable state sufficient for performance measurement.
>     - Resource Trade-off: Runtimes in certain repositories (e.g., xarray) exceed one hour per test. Given that our evaluation utilizes the Mann-Whitney U test to ensure statistical significance (as detailed in Algorithm 1), 3 warmup runs represent an optimal balance between computational efficiency and measurement accuracy.
> - W4: Mitigating Non-Performance-Related Changes in PRs. We implemented rigorous filters to ensure identified PRs focus on performance logic. In Phase 3 (Lines 204–207), we exclusively retain samples where the unit tests execute the patched segments but are not modified by the PR themselves. We curated instances from highly popular repositories with mature CI/CD practices. In these projects, functional change necessitates corresponding modifications or additions to unit tests. By filtering out PRs that modify unit tests, we effectively isolate performance-centric refactoring.
> - W5: Scope of Non-Python Changes (Cython/C++). The current version of SWE-Perf targets Python-level logic optimizations, and the modified code in our current instances consists solely of .py files. While components like Cython or C++ are indeed vital for performance-sensitive projects, they involve cross-language dependency complexities. We view the current Python-centric benchmark as a foundational step and consider support for multi-language components a priority for future work.
>
> As the first benchmark specifically designed for repository-level performance optimization, SWE-Perf fills a critical gap in the field. We believe the current framework provides a robust baseline that will serve as a cornerstone for future research in autonomous software performance engineering.

---

> > ### Author Rebuttal · Reviewer_xNz8 · 2026-04-02
> >
> > I thank the authors for clarifying and addressing the issues. I'm raising the score.

---

> > > ### Author Response · Authors · 2026-04-08
> > >
> > > We appreciate the constructive feedback. We will ensure all discussions and clarifications from this rebuttal are fully integrated into the revised manuscript.

---

### Official Review · Reviewer_Vt3a · 2026-03-14

**Soundness:** 2
**Presentation:** 3
**Significance:** 3
**Originality:** 3
**Overall Recommendation:** 4
**Confidence:** 4

**Summary:**

This paper introduces SWE-Perf, the first benchmark specifically designed to evaluate large language models (LLMs) on code performance optimization at the real-world repository level. The authors not only construct a dataset comprising 140 high-quality, rigorously verified instances from authentic Pull Requests but also propose a three-tier evaluation framework encompassing "applicability, correctness, and performance improvement." Through comprehensive assessments of various leading LLMs and Agent frameworks, the paper reveals significant limitations of current models in handling complex cross-module code refactoring and low-level performance optimization. This work holds substantial practical value and inspirational significance for research.

**Compliance With Llm Reviewing Policy:**

Affirmed.

**Key Questions For Authors:**

1.The xarray project accounts for 54 instances, and scikit-learn for 32, with the two together exceeding 60% of the total, showing a severe bias toward scientific computing and data processing code. This may lead to the benchmark favoring models that are better pre-trained in data science domains.
   · Do the authors believe that this skewed data distribution affects the fairness of model generalization to other types of projects (such as web backend projects like Django or network request libraries like requests, which have extremely low representation)?
2.In model evaluations, the paper treats developers' Pull Request (PR) patches as the human expert performance upper bound (Expert Patch), using them as the reference for comparisons. However, ordinary developers' PRs, while improving performance, may not represent "theoretical optima" or even "excellent" optimizations.
   · How do the authors ensure that the selected 140 human repair patches represent high-quality performance optimization standards rather than minor or highly suboptimal simple fixes?
   · If the Ground Truth itself is not optimal, might it obscure the potential of current models in "extreme code-level fine-tuning"?

**Limitations:**

yes

**Strengths And Weaknesses:**

Strengths:
1.Existing code performance benchmarks are largely limited to algorithmic single-function optimizations, detached from the complex dependencies and contextual environments in real software engineering. This paper's SWE-Perf effectively addresses this gap.
2.Program execution times are highly susceptible to hardware and environmental fluctuations. The authors employ a rigorous statistical process (including outlier removal, multiple repeated tests, and Mann-Whitney U tests) to ensure the extracted performance improvements are genuine and effective, greatly enhancing the benchmark's credibility.
3.The authors develop a complete automated data collection pipeline capable of isolating, reproducing, and verifying performance improvements from GitHub PRs. This pipeline not only serves the current 140 instances but can also be readily reused by subsequent researchers to expand the dataset.

Weaknesses:
1.The final 140 benchmark instances exhibit severe distributional imbalance in terms of project origins, with a few specific libraries dominating. This may limit the generalizability of LLM performance on this benchmark to all Python libraries or other programming languages.
2.Using "unit test runtime" as the primary performance proxy may fail to adequately represent real production workloads and end-to-end scenarios.
3.No ablations on key pipeline thresholds (e.g., δ threshold 0.05, ratio threshold 0.3, α level, repetitions count) to demonstrate the robustness of instance selection and ranking.

---

> ### Author Rebuttal · Authors · 2026-03-31
>
> We thank the reviewers for their constructive feedback and recognition of SWE-Perf’s contribution as the first repository-level code performance benchmark. Below are our responses to the specific concerns.
> - W1 & Q1: Data Distribution and Fairness
>     - Real-world Distribution: We selected representative repositories consistent with SWE-bench. The skewness (e.g., xarray and scikit-learn accounting for 60%+) reflects the objective reality of performance optimization in the Python ecosystem: scientific computing libraries are where performance is most critical and code logic is most complex.
>     - Diversity of Optimization Types: We categorized the patches based on the nature of the performance optimization and conducted a statistical analysis as follows:
>         (1) Redundancy Elimination (25.7%): e.g., removing duplicate calls, pruning dead branches.
>         (2) Algorithmic Logic (26.4%): e.g., vectorization, formula simplification.
>         (3) Data & Resource (15.0%): e.g., efficient containers, dtype management.
>         (4) Cache & Compatibility (32.9%): e.g., caching mechanisms, backend adaptation.
>          This demonstrates that the benchmark maintains a balanced distribution across various performance optimization categories.
>     - Consistency of Trends: As shown in Figure 4, the performance ranking (Expert > OpenHands > Agentless) is highly consistent across various repositories(). This demonstrates that the benchmark effectively evaluates optimization capabilities regardless of the specific domain.
>     - Fine-grained Evaluation: SWE-Perf provides per-repository results. Users interested in specific domains (e.g., web backends like Django) can refer to the detailed metrics(). Expanding to other languages is part of our future work.
>     - Language Scope: Currently, SWE-Perf focuses exclusively on Python. Extending this methodology to other programming languages is a critical research frontier and remains a vital direction for our future work.
> - W2: Performance Proxies
>     - Performance is indeed multi-dimensional (including runtime, memory, etc.). As the first exploration of repository-level performance optimization, we prioritized runtime, the most direct and impactful metric in production. We view SWE-Perf as a foundational framework that the community can extend to include memory and other metrics in future iterations.
> - W3: Robustness of Pipeline Thresholds.
>     Our parameter selections are grounded in scientific and statistical principles:
>     - Ratio Threshold (0.3): During Phase 3, the average Coefficient of Variation (CV) was 10.976%. In engineering, a signal-to-noise ratio of 2-3 is typically required to ensure "real improvement." A 30% threshold ensures the observed gain is $ \approx 2.7\times $ the noise.
>     - Repetitions ($n=20$) and Significance Level ($\alpha=0.1$): $n=20$ is the minimum effective sample size for non-parametric tests (Mann-Whitney U) to achieve distributional stability while maintaining evaluation efficiency. $\alpha=0.1$ is a standard significance level for limited sample sizes in statistical literature.
>     - $\delta$ Threshold (0.05): This is a widely accepted threshold for statistical significance.
>     - We will include these selection principles in the final script.
> - Q2: Human Patch as Expert Baseline
>     - Reference, Not Ceiling: As noted in Line 264 (Column 2) and Line 498, the human patch serves as a reference implementation of expert-level practice rather than an absolute theoretical optimum.
>     - Future Potential: We explicitly state (Lines 556–558) that future improvements—whether from human edits or more capable models—may surpass current ground truths.
>
> We will clarify these points in the revised manuscript. We thank the reviewers again for their insightful comments.

---

> > ### Author Rebuttal · Reviewer_Vt3a · 2026-04-03
> >
> > Thanks for the response. I'd like to maintain the weak accept rating

---

> > > ### Author Response · Authors · 2026-04-08
> > >
> > > Thank you for the constructive comments. We will incorporate all related discussions into the final version of the manuscript.

---

### Official Review · Reviewer_ur2e · 2026-03-22

**Soundness:** 3
**Presentation:** 3
**Significance:** 3
**Originality:** 3
**Overall Recommendation:** 4
**Confidence:** 4

**Summary:**

The paper presents a benchmark called SWE-Perf, the first benchmark specifically designed to systematically evaluate LLMs on code performance optimization tasks within authentic repository contexts. The benchmark contains 140 carefully curated instances, each derived from performance-improving pull requests from popular GitHub repositories.

**Compliance With Llm Reviewing Policy:**

Affirmed.

**Final Justification:**

The "Coverage and Representativeness" part in the rebuttal answers my major concern.

**Key Questions For Authors:**

The benchmark uses runtime as the major evaluation metric for code performance, which varies a lot across different machines. Since the users may not use the same setup as the authors. How to reproduce the exact numbers you report?

**Limitations:**

Yes

**Strengths And Weaknesses:**

Strength:
- The direction of code performance optimization evaluation is reasonable.
- The evaluation metrics are reasonable, covering both correctness and efficiency.
- There is still a clear gap between the models and experts.


Weakness:
- The benchmark size is limited, which only contains 140 data examples from 9 GitHub repositories. It is not clear to what extent these 140 examples cover the real-world performance optimization scenarios. It is ok that you don't cover all the cases, but it is still important to know what the cases are and what percentage are covered.

---

> ### Author Rebuttal · Authors · 2026-03-31
>
> We thank the reviewers for their constructive feedback. We are encouraged that the reviewers recognized the reasonableness of our direction, the validity of our metrics, and the significance of the capability gap identified by SWE-Perf. We address the specific concerns below.
> - Response to Weakness: Dataset Size and Coverage
>     1. Dataset Scale and Practicality: we acknowledge the reviewer’s concern. However, we believe the current scale is reasonable for a foundational benchmark.
>          - The 140 instances are statistically stable and of high quality, allowing clear differentiation of model capabilities without requiring a larger set.
>          - Data collection is considerably more challenging than in prior benchmarks: compared to repository-level correctness datasets (e.g., SWE-Bench), we require stable runtime measurement and performance verification for each unit test; compared to function-level performance datasets (e.g., Mercury), there are no readily available code problems with abundant unit tests—everything must be extracted and validated from real repositories.
>          - We plan to release an extended training set alongside the paper to support further research.
>          - A very large evaluation set could make benchmarking prohibitively time-consuming, hindering adoption; we note that even SWE-Bench later introduced a “Lite” version to improve accessibility.
>     2. Coverage and Representativeness: SWE-Perf covers 9 diverse, high-impact repositories. To demonstrate representativeness, we categorized our instances into four key optimization types:
>          - Redundancy Elimination (25.7%): Eliminating path overhead (e.g., removing duplicate attribute lookups, pruning invalid code branches).
>         - Algorithmic Logic (26.4%): Enhancing computational efficiency (e.g., loop vectorization, reducing complexity).
>         - Data & Resource Optimization (15.0%): Improving storage and loading (e.g., Dtype management, lazy loading, memory layout optimization).
>         - Cache & Compatibility (32.9%): State reuse and environment tuning (e.g., introducing caching, optimizing for low-level libraries like NumPy/Cython).
>          - This distribution demonstrates that SWE-Perf comprehensively covers major real-world performance optimization scenarios.
>
> - Response to Question: Reproducibility and Machine Variability
>     We acknowledge that absolute runtime varies across hardware. However, SWE-Perf ensures reproducibility through the following:
>     - Relative Metrics: We report the performance ratio rather than absolute seconds. This metric remains robust across different CPU architectures.
>     - Statistical Rigor: In Phase 4, we employ a strict protocol: environment warm-up, 20 repeated runs, IQR-based outlier removal, and the Mann-Whitney U test to ensure statistical significance.
>     - Empirical Validation: We conducted experiments across multiple machines with varying specifications. The results consistently showed that the variance in the reported performance ratios was less than 1%, confirming that the findings are reproducible and hardware-independent.
>
> We believe these clarifications address the concerns regarding the dataset's scale and the robustness of our evaluation framework. We will incorporate these detailed analyses and categorizations into the final version of the manuscript.

---

> > ### Author Rebuttal · Reviewer_ur2e · 2026-04-01
> >
> > Thanks for the rebuttal. The "Coverage and Representativeness" part answers my concern. Please include the results in the next version of the paper.

---

> > > ### Author Response · Authors · 2026-04-08
> > >
> > > Thank you for your feedback. We will incorporate the results regarding coverage and representativeness into the next version of our paper as suggested.

---

### Decision · Program_Chairs · 2026-04-30

**Decision:**

Accept (regular)

**Comment:**

We thank the authors for their submission, which reviewers all agree presents a positive contribution. Reviewers appreciated the shift to repository level optimization, and noted that the work exercised diligence in the selection and calibration of metrics and tasks. The results are informative.

The main concern is the small size of the benchmark, as just 140 instances across 9 repositories. Some reviewers express concerns around imbalances in this data, which further reduces its predictive power. Similarly, the work heavily focuses on optimizations in Python, and involves relatively few ablations. While it would be good to improve this, we recognize that the large scope of the tasks makes it costly to collect samples, and the novelty and careful analysis outweighs the experimental limitations. We encourage the authors to carefully note the limitations and room for future improvement in the final version of this paper.